# Use of Insecticides in Agriculture and the Prevention of Vector-Borne Diseases: Population Knowledge, Attitudes, Practices and Beliefs in Elibou, South Côte d’Ivoire

**DOI:** 10.3390/tropicalmed5010036

**Published:** 2020-03-01

**Authors:** Bédjou P. N’Dri, Kathrin Heitz-Tokpa, Mouhamadou Chouaïbou, Giovanna Raso, Amoin J. Koffi, Jean T. Coulibaly, Richard B. Yapi, Pie Müller, Jürg Utzinger

**Affiliations:** 1Swiss Tropical and Public Health Institute, P.O. Box, CH-4002 Basel, Switzerland; giovanna.raso@promotionsante.ch (G.R.); couljeanvae@yahoo.fr (J.T.C.); pie.mueller@swisstph.ch (P.M.); juerg.utzinger@swisstph.ch (J.U.); 2Department of Epidemiology and Public Health, University of Basel, P.O. Box, CH-4003 Basel, Switzerland; kathrin.heitz-tokpa@csrs.ci; 3Centre Suisse de Recherches Scientifiques en Côte d’Ivoire, 01 BP 1303 Abidjan 01, Côte d’Ivoire; mouhamadou.chouaibou@csrs.ci (M.C.); koffiamoinjeanned39arc@yahoo.fr (A.J.K.); richard.yapi@csrs.ci (R.B.Y.); 4Department of Entomology and Plant Pathology, North Carolina State University, Raleigh, NC 27695-7508, USA; 5Unité de Formation et de Recherche Biosciences, Université Félix Houphouët-Boigny, 22 BP 770 Abidjan 22, Côte d’Ivoire; 6Centre d’Entomologie Médicale et Vétérinaire, Université Alassane Ouattara, Bouaké, BP V 18 Bouaké 01, Côte d’Ivoire

**Keywords:** attitude, belief, Côte d’Ivoire, insecticide resistance, knowledge, malaria, practice, resilience

## Abstract

People’s knowledge, attitudes, practices and beliefs (KAPB) pertaining to malaria are generally well described. However, little is known about population knowledge and awareness of insecticide resistance in malaria vectors. The aim of this study was to investigate KAPB related to insecticide resistance in malaria vectors due to the use of insecticides in agriculture and the prevention against mosquitoes. In mid-2017, we carried out a cross-sectional survey in Elibou, South Côte d’Ivoire, employing a mixed methods approach. Quantitative data were obtained with a questionnaire addressed to household heads. Interviews were conducted with key opinion leaders, including village chiefs, traditional healers, heads of health centres and pesticide sellers. Focus group discussions were conducted with youth and elders. A total of 203 individuals participated in the questionnaire survey (132 males, 65%). We found that people had good knowledge about malaria and mosquitoes transmitting the disease, while they felt that preventing measures were ineffective. Pesticides were intensively used by farmers, mainly during the rainy season. Among the pesticides used, insecticides and herbicides were most commonly used. While there was poor knowledge about resistance, the interviewees stated that insecticides were not killing the mosquitoes anymore. The main reason given was that insecticides were diluted by the manufacturers as a marketing strategy to sell larger quantities. More than a third of the farmers used agricultural pesticides for domestic purposes to kill weeds or mosquitoes. We observed a misuse of pesticides among farmers, explained by the lack of specific training. In the community, long-lasting insecticidal nets were the most common preventive measure against malaria, followed by mosquito coils and insecticide sprays. The interviewees felt that the most effective way of dealing with insecticide resistance was to combine at least two preventive measures. In conclusion, population attitudes and practices related to insecticides used in agriculture and the prevention against mosquitoes could lead to resistance in malaria vectors, while people’s knowledge about insecticide resistance was limited. There is a need to raise awareness in communities about the presence of resistance in malaria vectors and to involve them in resistance management.

## 1. Introduction

Malaria remains a major public health problem with an estimated 228 million clinical cases and 405,000 deaths in 2018 [1]. Nevertheless, in the past 20 years, considerable progress has been made in malaria control by using long-lasting insecticidal nets (LLINs) and indoor residual spraying (IRS) [2,3,4]. Unfortunately, this progress is threatened by the development and spread of insecticide resistance in mosquitoes. Indeed, insecticide resistance is increasing throughout Africa, where most of the malaria burden is concentrated [5,6,7,8,9]. Due to practices linked to intensification of agriculture, resistance to insecticides has emerged and spread [10]. Several studies suggest that intensive use of insecticides in agriculture selects for resistant genes in malaria vectors [11,12]. Although research has been conducted previously on farmers’ knowledge, attitudes, practices and beliefs (KAPB) on insecticides used in agriculture [10,12], there is a paucity of studies on people’s knowledge about resistance to currently utilized insecticides and the causes leading to resistance, including how people cope with the growing resistance.

According to the World Health Organization, malaria is endemic in Côte d’Ivoire and the entire population is at risk of malaria transmission [1]. The main preventive measure is sleeping under LLINs [13]. Unfortunately, resistance to pyrethroids, which is the single class of insecticides currently approved for net impregnation, is widespread in Africa [14,15]. Additionally, pyrethroid insecticides are widely used in agriculture, i.e., for rice and vegetable farming [10]. Recently, an entomological study reported a high level of resistance to insecticides in *Anopheles gambiae* s.l. collected in Elibou in the southern part of Côte d’Ivoire [16]. Elibou is characterized by the absence of irrigated rice farming; other crops are cultivated at a distance of 2.5 km away from the village. The following research question was asked: what are the key practices of the population that might govern insecticide resistance?

To identify the potential root causes driving insecticide resistance in malaria vectors in Elibou, we carried out a KAPB survey using a mixed methods approach. We employed a household questionnaire to determine socio-demographic factors, population KAPB regarding the use of insecticides in agriculture and public health and resilience to resistance. In addition, we interviewed key informants, had focus group discussions (FGDs) and made direct observations.

## 2. Materials and Methods

### 2.1. Ethics Statement

The current study was approved by the ethics committees of North-Western and Central Switzerland (reference, 2017-00421 EKNZ) and the Comité National d’Éthique des Sciences de la Vie et de la Santé de Côte d’Ivoire (reference, 054/MSHP/CNER-kp). The head of Sikensi health district and village authorities gave permission to conduct the survey. The study procedures have been explained in lay terms to all participants, including objectives, potential risks and benefits related to the study.

### 2.2. Study Area

The study was designed as a cross-sectional survey. It was carried out in May and June 2017 in Elibou, a forested area located in the southern part of Côte d´Ivoire (geographical coordinates: 5°40’57.99” N latitude and 4°0’30.29” W longitude). Elibou is characterized by a long rainy season (April–July) and a long dry season (December–March), interspaced by a short rainy season (October – November) and a short dry season (August–September) (see: https://fr.climate-data.org; accessed on 6 January 2020). There is a swamp area with an environment suitable for mosquito breeding and larval development. Subsistence agriculture and trading are the main economic activities of the population. Cocoa, rubber tree (hevea), palm oil, tomato, chilly and cassava are the main crops.

### 2.3. Sample Size Calculation and Selection of Participants

We wanted to be 95% certain that the relative estimation error would not exceed 20% of the true proportion *p*, provided that *p* equals 1/3. This led to Equation (1):(1)$$0.2\times \frac{1}{3}\;\geq \;1.96\sqrt{\left(1/3\times 2/3\right)/n}$$
providing *n* > 192. Hence, we aimed to draw a random sample of at least 200 households.

If the head of a selected household was absent during the survey, he/she was replaced by someone from the household aged 20 years and above. If no such household member could be found, the neighboring household was chosen instead.

### 2.4. Questionnaire and FGDs

A KAPB survey was conducted with an emphasis on people’s use of insecticides in agriculture and for prevention of vector-borne diseases in the context of resistance in malaria vectors. The questionnaire was addressed to 203 heads of household and was structured around (i) socio-economic and demographic aspects of households; (ii) knowledge related to malaria and mosquito resistance; (iii) attitude, practices and beliefs related to the use of insecticide in agriculture and for disease prevention; and (iv) resilience to resistance of mosquitoes. Households were visited in the morning. A catch-up visit was carried out for those who were absent during our initial visits.

Four FGDs were carried out with elders and youth, separate by gender, each including eight individuals. Key informant interviews were conducted with five individuals (i.e., village chief, a pesticide seller, an elder person, a traditional healer and the head of the health centre). Two interview guides were employed; one for FGDs and one for key informant interviews.

Pesticides were defined as a substance used to control pests considered being harmful for humans and agricultural production, including weeds. There are three groups of agricultural pesticides: (i) insecticides used to fight insects; (ii) herbicides against weeds that impact the growth of plant; and (iii) fungicides against parasitic fungi that cause various diseases in plants.

### 2.5. Statistical Analysis

Data from the questionnaire survey were double entered into EpiInfo version 3.5.3 (Centers for Disease Control and Prevention; Atlanta, GA, USA) and cross-checked. The cleaned dataset was transferred into STATA IC 14 (Stata Corporation; College Station, TX, USA). Results are reported as counts and percentages. Chi-square (χ^2^) test was used to compare proportions between groups. Key informant interviews and FGDs were tape-recorded and transcribed in Word and converted into MAXQDA version 10 (VERBI Software Consult; Berlin, Germany) for qualitative data analysis and interpretation.

## 3. Results

### 3.1. Characteristics of Study Population

The socio-economic and demographic profile of the 203 respondents is summarized in Table 1. Most of the respondents were aged between 25 and 49 years (60.1%). With regards to education, slightly more than a quarter never attended school, while more than four out of 10 respondents only had obtained primary schooling (42.4%). Almost half of the respondents were famers (48.3%).

In Côte d’Ivoire’s multi-ethnic agrarian societies, the population is generally categorized into three groups of origin. In French, these are known as (i) *autochtone*s (indigenous; first settler population); (ii) *allochtones* (allochthonous; internal or national migrants, people stemming from another region of Côte d’Ivoire); and (iii) *allogènes* (allogenic; non-national migrants and foreigners). The *autochtones* of Elibou belong to the “Abidji’’, which is the predominant ethno-linguistic group (71.9%).

### 3.2. Awareness Related to Malaria and Insecticide Resistance

Most of the respondents (93.1%) complained about mosquitoes. Key informant interviews and FGDs suggested that the abundance of mosquitoes had increased over time. For instance, the village chief said: “*In the past, we did not feel the presence of mosquitoes or if they were there, they did not do so much damage*”. By the “past”, respondents referred to the 1980s and 1990s. During that time, the village chief and other elderly interviewees said that aerial sprayings were performed every couple of months to fight against onchocerciasis and human African trypanosomiasis. According to the village authorities, the spraying had the effect that the population did not feel the presence of mosquitoes and other disturbing insects. However, around the new millennium, the situation changed. Aerial sprayings were stopped and people began complaining about the large abundance of mosquitoes.

Regarding seasonal patterns, 61.1% of respondents stated that mosquitoes were most abundant during the rainy season. Only 9.9% stated that the highest abundance of mosquitoes occurs in the dry season. The remaining 29.1% gave no peak time of mosquito abundance. Insalubrity and backwater were the most frequently reported causes of mosquito abundance in the village, as illustrated by a quote from an elderly man: “*In former times, the village was maintained, and we did not have polluted water like this*. *There was a corner where we put waste. We did not have open wells, so there were fewer mosquitoes*’’. The fact that the village is no longer well maintained was explained by the demographic expansion and the arrival of migrants during the political crisis of 2002−2011. Some Christians interviewed referred the negative changes to prophecies in the Bible. According to a young man: *“The abundance and resistance in mosquito is a punishment of God. These are warning signs of the end of the world”*. Finally, people linked the mango and maize season—which correspond to the long rainy season—to the presence of large numbers of mosquitoes.

Most respondents (98%) answered having good knowledge about malaria with no statistically significant difference according to educational attainment (χ^2^ = 1.8; *p* = 0.6). In the local language (i.e., Abidji), malaria is termed “*djèkouadjo*”. There are different types of *djèkouadjo*. For instance, the traditional healer distinguished between two types of malaria (i) “*djèkouadjo lébé*”, which is the yellow malaria and (ii) “*djèkouadjo lofou*”, which means white malaria. The latter is more dangerous and can cause madness.

Respondents mentioned that mosquitoes are responsible for diseases, such as malaria, typhoid fever and anaemia. Indeed, 94.1% of the respondents knew that malaria is caused by mosquito bites. The results revealed that malaria was not so common in earlier times, and people found it easier to treat with traditional remedies. To date, residents still rely on traditional medicine. They only go to the hospital when the disease has become severe. According to the head of the local health centre: “*[People] go to the traditional healer and, when they see that it’s complicated, they send the child here. You see the child in a comatose state*”.

Table 2 summarises people’s knowledge about insecticide resistance in mosquitoes. Regarding knowledge of insecticide resistance in the malaria vector, a similar percentage of respondents felt that insecticides are effective (42.9%) or ineffective (44.3%), while the remaining 12.8% had no clear opinion. Hence, if one exclusively takes this response into consideration, it is not possible to determine whether insecticides are, or have become, ineffective. Nevertheless, among those who perceived a decline in the effectiveness of insecticides, 57.8% stated that the root cause was the dilution of insecticides by the vendors.

For some participants it is not conceivable why the same insecticides that effectively killed mosquitoes in the past are no longer effective today. Other participants see a link between insecticide resistance and the business strategy of the insecticide manufacturers and vendors. An elderly man explained during a FGD: “*Manufacturers do not dose insecticides well. They know that if they dose it well and (the product) kills mosquitoes, we won’t buy their products any longer. So they have reduced the dose to continue selling their products*”. Yet for others, the fact that it is the same treatment used in the past and today proves that mosquitoes have become resistant to the products used. This observation was articulated as follows: *“The way our organism adapts, this is how mosquitoes have changed, too. So I think if the treatments don’t kill them anymore, it is because their organism have become used to the same treatment”.*

### 3.3. Attitudes, Practices and Beliefs Related to Insecticide Use in Agriculture and Disease Prevention

Table 3 illustrates attitudes and practices of participants in relation to insecticides used while pursuing agricultural activities. Three quarter of the respondents have a farm (n = 152, 74.9%). Among them, 75% use pesticides. Farmers mentioned that the choice of pesticides they use depends on their crops. A seller of pesticides explained during the interview: “*There are several types of products; there is one for pepper, okra, eggplant, rubber and cocoa. Some people also use them to kill insects that attack plants, others on the contrary use them to weed their fields*”. The respondents’ believe that using the products will increase their production, and hence their income. The rainy season is the period when respondents are most likely to buy and apply insecticides and herbicides. As quoted by a pesticide seller: “*It is in the rainy season that things move. During this period, the people use a lot of herbicides*”.

As shown in Table 3, most farmers use agricultural pesticides in their homes against weeds and for mosquito control (39.1%). It is important to note that the same pesticides mentioned are used at home. A famer in a FGD said: *“When we spray the products, it kills the weeds and all the insects that are inside. There are other products that repel and kill snakes, so it is unlikely that the mosquitoes will stay alive. For us, the effect that these products can have on mosquitoes is that it kills them”.* Moreover, the use of herbicides for weeding allows saving time and energy, as illustrated by a man in his 40s: *“We prefer to take a bottle of herbicides to spray the grass around the house, it is fast and then it is less tiring than taking a machete or hoe”*.

Farmers interviewed have the habit of using agricultural pesticides for mosquito control and weeding at home without considering the risks that this might have on their health. One of them, a household head, confirmed to usually spray *“decis”* insecticide in his room due to the fact that mosquitoes are resistant to other preventive measures. Regarding farmers’ educational attainment, 29% who used pesticides were illiterate, 40% only attended primary school, 26.3% secondary school and 5.3% had high school education.

Insecticides, herbicides, fungicides and growth regulators were mentioned during the cross-sectional survey. Insecticides were frequently used by farmers (45.7%), followed by herbicides (40.0%). Fungicides and growth regulator accounted for 8.6% and 5.7%, respectively. Among insecticides used, pyrethroids (e.g., deltamethrin, cypermethrin, D-althrin, bifenthrin and lambdacyalothrin) accounted for 57.1%. Neonicotinoids (e.g., acetamiprid and thiamethoxin; 28.6%) and organophosphates (e.g., etonophos and profenofos; 14.3%) were additional important classes of insecticides.

Regarding different preventive measures used by those who perceived a change in the effectiveness of insecticides, LLINs were most often stated (25.6%). Yet, the actual use of LLINs was low. Reasons given for not using LLINs are that sleeping under a LLIN causes discomfort by suffocation and excess heat (18.2%) or that people did not receive LLINs (18.0%). The use of fumigating coils emerged as another important tool to protect against mosquito bites, as stated by 15.3% of the respondents. In practice, most of the participants in the FGDs and those interviewed, used fumigant coils with the commercial name *“moustico”*. This device is considered as a cheaper option compared to LLINs. Its use is not only price-related but also by habits. A farmer in a FGD explained: *“We don’t have money, so it is “moustico” that we use. Since we were children, it is what our parents used to repel mosquitoes.”*

Slightly more than half of the participants (53.2%) used a single tool to prevent mosquitoes (Table 4). With regard to using multiple preventive measures, the combination of LLINs and IRS was most often given (19.7%). A man in his 40s said: *“When we use one insecticide and mosquitoes are still there, we start using both insecticide spray and fumigant coils at the same time, so that, the dose of insecticides becomes strong and we get to sleep a little.”*

Even though people use insecticide sprays, the reported quantity is small. During our FGDs, participants reported not to spray in all the rooms and only to spray a small amount, because the product was too expensive. Respondents believe that it is the smell of the product that repels mosquitoes. This practice is a strategy to make savings, given that the spray is quite expensive.

Taken together, there is a good management of insecticide sprays used. An elderly man explained: “*I use insecticide spray, so in the evening, we spray at least 30 minutes before sleeping. Moreover, we spray a little bit because it costs FCFA 1,000, we cannot buy it each week. The smell is enough; this is why we spray just a little bit*.” This attitude from respondents could also be linked to the fact that participants cannot afford to spray frequently at large quantities. This is illustrated by a woman in her 50s: “*Me, I used to use insecticide spray. But when I saw that mosquitoes were coming a lot, that’s when I stopped, because I do not have money*”.

### 3.4. Population Resilience Regarding Resistance to Insecticides

According to the results from the FGDs and interviews, the main tool to fight against mosquito bites in the past was by burning the bark of a tree, locally known as “*gnanman*”. The traditional healer of the village said: “*At the time, there was a bark of a tree which served to repel mosquitoes and this tree is called gnaman. Now it does no longer exist due to the production of coal and the destruction of the forest*”.

For the majority of respondents, alternative strategies consist of using a bed sheet to cover themselves or a piece of cloth to repel mosquitoes. Although they developed strategies to avoid mosquito bites, 55.2% of respondents ranked LLINs as the most effective tool. Participants in FGDs said that they used LLINs when they noticed the limitations of the above strategies. This is illustrated by the following quote from an elderly man: *“After we have tried everything and it did not work finally, we used mosquito nets. But again, it did not work, because even being under the net, we still had mosquito bites”.* Beside the use of LLINs, respondents have adopted additional strategies such as environmental sanitation as the most important one, as stated by 72.4% of the respondents.

Results from the FGDs and interviews revealed that participants keep their surrounding clean. Key activities are focused on weeding and cleaning the compound. However, these activities were limited to the immediate surrounding of participants’ homes and not the public space of the village. A woman in her 60s said in a FGD: *“Well, to avoid mosquitoes, I clean the house, I sweep the room. But without effect, mosquitoes are still there”.*

Regarding the options given for enhanced control, respondents have identified people who should be involved in vector control activities in Elibou. Respondents felt that the following actors should deal with vector control: the general population (47.8%), municipality (22.7%) and traditional authorities (e.g., village chief and other local authorities; 12.8%). However, about one of seven participants did not know who should deal with vector control. FGDs and interviews revealed that everyone has to promote environmental sanitation in his or her neighbourhood*: “I propose that we group young men and women by neighbourhood. Specifically, clean, sweep [and] empty garbage. We can collect garbage, because it is a source of mosquito production”.*

## 4. Discussion

In the current study, 39% of respondents confirmed to have a habit of using agricultural pesticides in their houses against weeds. Among them, one household head affirmed to spray “*decis*” in the rooms due to the ineffectiveness of disease prevention tools commonly used against mosquitoes. While this insecticide belongs to the class of the pyrethroids, it is intended for treatments of various plant cultures. The use of insecticides in agriculture and for disease prevention is an important cause for the development and spread of resistance in mosquitoes. Indeed, the spread of insecticide resistance is a public health threat, which can jeopardize vector-borne disease control efforts. In Côte d’Ivoire, resistance to insecticides is widespread, presumably as a result of the heavy use of pyrethroids in agriculture [10,17,18]. In 1993, resistance to pyrethroids was reported for the first time from Bouaké in the central part of Côte d’Ivoire [18]. More recently, a high level of resistance to insecticides in malaria vectors has been reported from Elibou where the current study was conducted [16]. Awareness of resistance in mosquitoes was less known by respondents. A deeper understanding of the population’s KAPBs in a given community regarding resistance in malaria vectors might help improve control strategies [19].

Our results revealed that the same insecticides used in agriculture are also employed at home against weeds and for mosquito control. About 75% of field owners in the current study used pesticides in agriculture. Thus, the concomitant use of insecticides in agriculture and for disease prevention are likely causes for the observed resistance to insecticides in malaria vectors in Elibou [16]. Moreover, the aerial spraying in the last decades before the new millennium, using DDT, organophosphates, carbamates and pyrethroids against black flies and tsetse flies, as reported during our FGDs, might also explain the selection for resistance in mosquitoes. Considering that all LLINs contain pyrethroids, our observations are worrying with regards to vector-borne disease control and the potential re-emergence and persistence of malaria. Several studies emphasized the potential role of agricultural insecticides in accelerating the selection of insecticide resistance due to their excessive use [10,17,20,21].

Pesticides are primarily applied during the rainy season and might then be washed out by the rains and contaminate the village of Elibou. As mosquito breeding sites are abundant during the rainy season, the pesticides may also affect larvae breeding sites. The pesticide residues seep into the soil, likely exerting a selection pressure on the larvae [8]. Among the pesticides identified, insecticides and herbicides were used most commonly. Pyrethroids, neonicotinoids and organophosphates were the main classes of insecticide applied by farmers. Among them, pyrethroid insecticides were the principal class. The increased use of pyrethoids could also explain the high resistance to this class of insecticides observed in malaria vectors in Elibou. Moreover, most farmers have not received specialists’ advice on the best choice of a product and adequate use of pesticides, such as the exact dosage and frequency of use. Given that 29% of pesticides users are illiterate [10], they may also ignore label instructions. Finally, there is a lack of control of the quality of products that are used by the farmers for crop production leading to sales and use of pesticides not registered in Côte d’Ivoire.

In contrast to the general unawareness of how to properly use pesticides, most participants had good knowledge about malaria. Half of the participants perceived a decline in the effectiveness of malaria vector prevention tools over time. Hence, we conjecture that villagers have observed some kind of mosquito resistance development. Interestingly, 93% of participants noticed an increase in mosquito densities and half of them attributed the increase to the ineffectiveness of insecticides used to prevent mosquitoes. According to participants’ responses, the products sold on the Elibou market have been adulterated by manufacturers, which might explain the reduced efficacy. Participants attributed it to the business of manufacturers to increase the return on investment with fraudulent means. In fact, they cannot imagine that the same insecticides that effectively killed and repelled mosquitoes in the past do not work any longer. Hence, there is a pressing need to inform users on the current situation to raise awareness. As protective measure, LLINs remain the most effective tool for prevention of mosquito bites, and thus the transmission of malaria [4]. However, its usage is low due to the fact that people do not like the extra heat while sleeping under a LLIN or unavailability of LLINs, among other reasons. These factors constitute a weakness that impedes malaria control activities. Similar observations were made in previous studies conducted in Ghana, Nigeria and other parts of the world [22,23].

A follow-up of nets distributed, and their proper use is required. The majority of respondents have low socio-economic status that compromised their attitude and practices on the use of insecticides. Indeed, the use of prevention measures depend on the cost of the device. Education and specific training on indoor spray usage is required because participants have the habit to spray only a little amount of insecticides in only a part of their rooms. Even though they might have a large house, they preferred to spray in the main room only. Actually, the amount sprayed was not enough to repel and kill mosquitoes throughout the house. This practice is also governed by the cost of the products. Fumigant coils are being used as an alternative to LLINs as they are thought to be cheaper. Moreover, participants used *“moustico”* not only because of its perceived low cost, but also out of habit. According to them, “*moustico*” has been used for a long time by their parents and therefore, they continue to use it despite its ineffectiveness against mosquitoes.

Owing to the ineffectiveness of the use of a single tool to protect against mosquito bites, respondents developed a more effective strategy, such as combining two or more preventive measures. Quite often, LLINs and insecticide sprays were used in conjunction, as it is believed that relying on both products simultaneously is more effective than a single tool. Several studies reported the significant effect to combine both LLINs and IRS usage on malaria, particularly in areas where pyrethroid resistance is high [2,24]. However, the rate of multiple tools observed in our study was still low, perhaps, because the spray is too expensive. Thus, the practice of covering oneself in a bed sheet to avoid mosquito bites at night appears as an additional solution to prevent from mosquito bites.

Vector control requires community engagement. Interestingly, participants have made some suggestions in order to reduce and eliminate resistance. First, the involvement of the population to promote cleanliness in neighbourhoods, which should be enforced by local authorities. We suggest regular monitoring and training of LLINs distributed by the national malaria control programme to their correct use, education of the community for the proper use of insecticides applied for agriculture and disease prevention and implementation of a rigorous control of pesticides sold to avoid the hazards of toxic products non authorized by the government of Côte d’Ivoire.

Only a few studies have been carried out on KAPB of populations on their habits related to the use of insecticides in agriculture and disease prevention, which could impact on insecticide resistance [10,17]. The current study is the first report on data of population knowledge about resistance in malaria vectors. Unlike malaria, we recorded that knowledge of the resistance phenomenon is not well understood by the community. Hence, there is a need to inform people about this issue. This is important, since the main users of pesticides for cultivation are illiterate or have only attained primary schooling. The use of insecticides in agriculture depends on the culture and farmers’ financial resources [17]. Similar observations were made in our study. Herbicides and insecticides were the pesticides most often used by the respondents. Our results corroborate with those found by Chouaïbou et al. [10] for vegetable cultivation. In Elibou, there is no irrigated rice farming in the main village. The nearest fields with other crops are located 2.5 km away. Our study also revealed that pesticides are more heavily used during the rainy season. As the rainy season is the period where mosquitoes are most abundant, it is essential that people use prevention tools to avoid mosquito bites. To sum up, the more frequently insecticides are being used, the higher the risk of selection and spread of resistance.

Our study has several limitations that are offered for consideration. First, the study focussed on a single village, due to constrained financial resources. Hence, generalizability to other parts of Côte d’Ivoire is currently not possible. Second, the cross-sectional study design impedes causal inference. A strength of our study was that the estimated sample size of 200 was attained, thanks to the enthusiastic participation of villagers. Indeed, we were able to enrol slightly more people who generously provided answers to all questions.

## 5. Conclusions

The simultaneous use of insecticides in agriculture and disease prevention is a driver for resistance in malaria vectors. Further studies are required to deepen the understanding of this issue, including investigations on insecticide residuals in water and soil in the village and surrounding farms. Additionally, in order to reduce the effect of pesticides used on the environment and people’s health, knowledge of proper handling and use are required. This is the first study investigating KAPB about insecticide resistance in Côte d’Ivoire. For the national malaria control programme, experiences and lessons should be utilized while developing and implementing a strategy for resistance management.

## Figures and Tables

**Table 1 tropicalmed-05-00036-t001:** Characteristics and socio-economic status of the study populations in Elibou, South Côte d’Ivoire in 2017 (n = 203).

Characteristic	No. of People Interviewed	Percentage
Sex		
Male	132	65.0
Female	71	35.0
Age group (years)		
20−24	12	5.9
25−49	122	60.1
≥ 50	69	34.0
Educational attainment		
None	53	26.1
Primary school	86	42.4
Secondary school	48	23.6
High school	16	7.9
Occupation		
Farmer	98	48.3
Housewife	50	24.6
Trader	16	7.9
Civil servant	14	6.9
Unemployed	25	12.3
Population group		
*Autochtones* (Abidji)	146	71.9
*Allochtones*	43	21.2
*Allogènes*	14	6.9
Wealth tertile		
Poorest	71	35.0
Less poor	65	32.0
Least poor	67	33.0

**Table 2 tropicalmed-05-00036-t002:** Responses given by study participants in relation to the knowledge of insecticide resistance in malaria vectors from a cross-sectional survey conducted in Elibou, South Côte d’Ivoire in 2017.

Question	n (%)
Are insecticides effective against mosquitoes? (n = 203)	
No	90 (44.3)
Yes	87 (42.9)
I do not know	26 (12.8)
What are the causes of ineffectiveness of insecticides? (n = 90) *	
Insecticides have been diluted	52 (57.8)
Mosquitoes have become numerous and dangerous	12 (13.3)
Due to vector control	4 (4.4)
I do not know	22 (24.4)
Since when have you observed that insecticides have been ineffective? (n = 90) *	
After regional vector control was stopped	28 (31.1)
Always	9 (10.0)
For more than 10 years	1 (1.1)
I do not know	52 (57.8)

* n = 90: number of participants who responded that insecticides were not effective.

**Table 3 tropicalmed-05-00036-t003:** Attitudes and practices of respondents from Elibou, South Côte d’Ivoire regarding the use of insecticides, as revealed by a cross-sectional survey in 2017.

Agricultural practices	n (%)
Do you have a farm? (n = 203)	
Yes	152 (74.9)
No	51 (25.1)
Do you use pesticide on your farm? (n = 152)	
Yes	115 (75.7)
No	37 (24.3)
Period of pesticides usage? (n = 115)	
All seasons	63 (54.8)
Rainy season	39 (33.9)
Dry season	13 (11.3)
Do you use agricultural insecticides in your home? (n = 115)	
Yes	45 (39.1)
No	70 (60.9)

**Table 4 tropicalmed-05-00036-t004:** Tools used for protection against mosquitoes mentioned by respondent from Elibou, South Côte d’Ivoire who perceived insecticides as inefficient.

Preventive Measure	n (%)
None	11 (5.4)
Single tool	
LLIN	52 (25.6)
Fumigant coil	31 (15.3)
Insecticides spray	13 (6.4)
Electric fan	1 (0.5)
Multiple tools	
LLINs + insecticide spray	40 (19.7)
LLINs + insecticide spray + fumigant coil	19 (9.4)
LLINs + fumigant coil	18 (8.9)
Insecticide spray + fumigant coil	15 (7.4)
LLINs + fan	2 (0.9)
Fumigant coil + fan	1 (0.5)

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
