# Peer review of "Use of Insecticides in Agriculture and the Prevention of Vector-Borne Diseases: Population Knowledge, Attitudes, Practices and Beliefs in Elibou, South Côte d’Ivoire"

_tropicalmed, 2020, doi:10.3390/tropicalmed5010036_

Round 1

Reviewer 1 Report

The article describes the knowledge, attitudes, practices, and beliefs (KAPB) of residents in Elibou, South Cote d'Ivoire to mosquito resistance to insecticides, both in spraying and LLINs. This manuscript presents the data found in an unbiased manner. Findings presented should be interesting to the readership of this journal.

Minor correction: line 164 educational attainment (spelling error).

Reviewer 2 Report

The study of N'Dri and colleagues is very important in regard to the current context of vector-borne diseases, particularly malaria, control. 

It helps to understand how population KAPB could contributes to the emergence of mosquitoes resistance to insecticides and their perceiption of the current vector control tools.

Minor remarks:

Lines 61-62: the authors stated that" their is no irrigated rice farming and no field around Elibou. But it is contradictory to what is described in lines 87-88

Table 1: how did the authors to evaluate the wealth of the population of study?

Is their any link between the level of wealth and the KAPB?  what is the influence of age on the KAPB?
